# Effects of Ethanol Extracts from *Grateloupia elliptica*, a Red Seaweed, and Its Chlorophyll Derivative on 3T3-L1 Adipocytes: Suppression of Lipid Accumulation through Downregulation of Adipogenic Protein Expression

**DOI:** 10.3390/md19020091

**Published:** 2021-02-04

**Authors:** Hyo-Geun Lee, Yu-An Lu, Jun-Geon Je, Thilina U. Jayawardena, Min-Cheol Kang, Seung-Hong Lee, Tae-Hee Kim, Dae-Sung Lee, Jeong-Min Lee, Mi-Jin Yim, Hyun-Soo Kim, You-Jin Jeon

**Affiliations:** 1Department of Marine Life Science, Jeju National University, Jeju 63243, Korea; hyogeunlee92@gmail.com (H.-G.L.); luyuan@jejunu.ac.kr (Y.-A.L.); wpwnsrjs@naver.com (J.-G.J.); tuduwaka@gmail.com (T.U.J.); 2Research Group of Food Processing, Korea Food Research Institute, 245, Nongsaengmyeong-ro, Iseo-myeon, Wanju 55365, Korea; mckang@kfri.re.kr; 3Department of Pharmaceutical Engineering, Soonchunhyang University, Asan-si 31538, Korea; seunghong0815@gmail.com; 4Naturetech Co., 29-8, Yongjeong-gil, Chopyeong-myeon, Jincheon 27858, Korea; taehee0317@naver.com; 5National Marine Biodiversity Institute of Korea, 75, Jangsan-ro 101-gil, Janghang-eup, Seocheon 33362, Korea; daesung@mabik.re.kr (D.-S.L.); lshjm@mabik.re.kr (J.-M.L.); mjyim@mabik.re.kr (M.-J.Y.)

**Keywords:** red seaweed, *Grateloupia elliptica*, obesity, adipocyte, adipogenesis

## Abstract

*Grateloupia elliptica* (*G. elliptica*) is a red seaweed with antioxidant, antidiabetic, anticancer, anti-inflammatory, and anticoagulant activities. However, the anti-obesity activity of *G. elliptica* has not been fully investigated. Therefore, the effect of *G. elliptica* ethanol extract on the suppression of intracellular lipid accumulation in 3T3-L1 cells by Oil Red O staining (ORO) was evaluated. Among the eight red seaweeds tested, *G. elliptica* 60% ethanol extract (GEE) exhibited the highest inhibition of lipid accumulation. GEE was the only extract to successfully suppress lipid accumulation among ethanol extracts from eight red seaweeds. In this study, we successfully isolated chlorophyll derivative (CD) from the ethyl acetate fraction (EA) of GEE by high-performance liquid chromatography and evaluated their inhibitory effect on intracellular lipid accumulation in 3T3-L1 adipocytes. CD significantly suppressed intracellular lipid accumulation. In addition, CD suppressed adipogenic protein expression such as sterol regulatory element-binding protein-1 (SREBP-1), peroxisome proliferator-activated receptor-γ (PPAR-γ), CCAAT/enhancer-binding protein-α (C/EBP-α), and fatty acid binding protein 4 (FABP4). Taken together, our results indicate that CD from GEE inhibits lipid accumulation by suppressing adipogenesis via the downregulation of adipogenic protein expressions in the differentiated adipocytes. Therefore, chlorophyll from *G. elliptica* has a beneficial effect on lipid metabolism and it could be utilized as a potential therapeutic agent for preventing obesity.

## 1. Introduction

Obesity has become the most prevalent nutritional problem globally. According to the World Health Organization (WHO), worldwide obesity has tripled since 1975 and was first recognized as a disease in 1948 [1]. In 2015, approximately 2.3 billion people were overweight and over 700 million people suffered from obesity around the world [2]. Obesity is associated with complicated metabolic diseases and characterized by high lipid content in the blood and adipocytes as well as inappropriate hormone secretion capacity from adipose tissues [3,4]. Obesity can be attributed to excessive food intake [5,6,7]. Long-lasting overweight and obese conditions induce negative effects on human health [8,9], such as obesity-associated metabolic diseases, including diabetes mellitus and cardiovascular disease [10,11,12,13]. Thus, several anti-obesity agents have been developed in the medicine and functional food industries. Currently, two available anti-obesity agents, orlistat and sibutramine, which are chemically synthesized, are used for the treatment of obesity [14,15,16]. Orlistat, a potent inhibitor of both pancreatic and other lipases, inhibits lipid metabolism and decreases lipid accumulation by inhibiting lipid absorption in the human intestine [17]. Sibutramine is an appetite suppressant that inhibits serotonin and norepinephrine reuptake and may also increase total serotonin levels in the brain [18]. Sibutramine reduces lipid storage and synthesis, resulting in body weight loss [19]. However, this synthetic anti-obesity agent has undesirable side effects. The most frequently reported side effects of sibutramine are gastrointestinal-related steatorrhea with excessive flatus [20] and a number of severe hepatic adverse effects (hepatic failure, necrosis, and cholestatic hepatitis) [21,22,23,24,25]. In some cases, sibutramine treatment causes anorexia, headache, insomnia, constipation, and dry mouth. In addition, long-term treatment with sibutramine can result in severe cardiovascular diseases including non-fatal myocardial infarction and stroke in obese patients [26]. Accordingly, the development of new natural anti-obesity agents is needed to treat obesity without inducing side effects.

Seaweeds contain numerous bioactive components and many red seaweed extracts (*Kappaphycus alvarezii*, *Gelidium amansii*, *Plocamium telfairiae*) have demonstrated anti-obesity properties under in vitro and in vivo conditions [27,28,29,30]. In this study, we investigated the potential lipid inhibitory effect of the active compound of red seaweed extracts on 3T3-L1 cells.

*Grateloupia elliptica* (*G. elliptica*) is a red seaweed present in the intertidal zone and distributed across Japan and Korea [31]. Previous studies have reported that the bioactive polyphenolic compounds from *G. elliptica* shows a potent antidiabetic activity [32] and that *G. elliptica* water extracts exhibit a preventative effect on hair loss in dermal papilla cells [33]. Additionally, a study on the chlorophyll demonstrated its potential anticancer activity in human glioblastoma cells [34]. However, there have been no reports on the lipid inhibitory effect of *G. elliptica* and its inhibitory effect on adipogenic protein expression in 3T3-L1 adipocytes. In this study, we investigated the potential lipid inhibitory effect of *Grateloupia elliptica* 60% ethanol extract (GEE) and its active compound, chlorophyll derivative (CD), on 3T3-L1 adipocytes in vitro by Oil Red O staining its mechanism by Western blot analysis.

## 2. Results

### 2.1. Effects of Red Seaweed Extract on Lipid Accumulation in 3T3-L1 Cells

In this study, the potential lipid inhibitory effect of eight red seaweeds were screened on 3T3-L1 differentiated adipocytes. Table 1 shows the list of eight red seaweeds. Figure 1A shows microscopic images of the 3T3-L1 cells treated with red seaweed ethanol extracts, and Figure 1B, C shows the cell viability and relative quantity of accumulated intracellular lipid contents in 3T3-L1 cells. All the tested samples did not exert cytotoxic effect and significant lipid inhibition was observed in *Grateloupia elliptica* (GE), *Grateloupia lanceolata* (GL), *Gracilaria verrucosa* (GV), *Gloiopeltis furcata* (GF) and *Lomentaria catenata* (LC) treated 3T3-L1 cells. Among the eight red seaweeds, the *Grateloupia elliptica* 60% ethanol extract (GEE) showed the highest inhibition of intracellular lipid accumulation in 3T3-L1 cells. In an additional experiment, GEE significantly reduced lipid accumulation in 3T3-L1 cells (Figure 2). Taken together, these results suggested that GEE had a potent lipid inhibitory effect on differentiated 3T3-L1 cells. Accordingly, *G. elliptica* was selected for further isolation of bioactive compounds.

### 2.2. Effects of Solvent Fractions from GEE on Intracellular Lipid Accumulation in 3T3-L1 Cells

The inhibitory effect on intracellular lipid accumulation of solvent fractions from GEE was evaluated by Oil Red O (ORO) staining. Figure 3A shows the microscopic images of 3T3-L1 cells and Figure 3B, C shows the cell viability and relative lipid content of 3T3-L1 cells treated with different concentrations (125, 250, and 500 μg/mL) of solvent fractions. According to the results, the solvent fractions did not have significant cytotoxicity on 3T3-L1 cells. In addition, the lipid accumulation was gradually decreased with increasing concentrations of most solvent fractions; among the solvent fractions, the ethyl acetate (EA) fraction reduced lipid accumulation more effectively than the other fractions. Therefore, the EA fraction was chosen as a target fraction as it exhibited significant inhibition of lipid accumulation in vitro. These results indicated that the EA fraction from GEE might contain the active compound, exhibiting a suppressive effect on lipid accumulation in 3T3-L1 cells and having potent anti-obesity activity.

### 2.3. High Performance Liquid Chromatography Analysis of the EA Fraction and GEE

In a previous experiment, ORO results demonstrated that the EA fraction shows an excellent inhibitory effect on intracellular lipid accumulation in 3T3-L1 cells. Accordingly, the HPLC analysis was carried out for GEE and its EA fraction to purify the active compound. The HPLC chromatograms of GEE and the EA fraction showed a similar pattern of HPLC peaks: one major peak (EA-a) was observed in the EA fraction (Figure 4). These results indicated that the EA-a peak was highly concentrated in the EA fraction during fractionation.

### 2.4. Identification of Chlorophyll Derivative via HPLC and Absorption Spectrum

The chlorophylls were identified using HPLC chromatogram and absorption spectrum analysis. Figure 5 shows the HPLC chromatogram of the main peak and the absorption spectrum at wavelengths between 200 and 700 nm. The HPLC chromatogram shows that the EA-a peak was detected at 400 and 667 nm. The absorption spectrum shows that the EA-a was detectable in the spectrum at wavelengths of 399 and 667 nm. It was previously reported that chlorophyll derivative (CD) is detected at specific absorption wavelengths of 400 and 667 nm [35,36]. The comparison of the spectrum of the major peak in the EA fraction implies that the major peak is CD.

### 2.5. Effect of Chlorophyll Derivative on Lipid Accumulation and the Expression of Adipogenic Proteins in 3T3-L1 Cells

The cytotoxic effect of chlorophyll derivative (CD) on 3T3-L1 cells was investigated to assess whether it could be commercially used as an anti-obesity agent. The toxicity of CD was determined by the 3-(4,5-dimethylthiazol-2-yl)-2,5-diphenyltetrazolium (MTT) assay. The 3T3-L1 cells were seeded in 48-well plates and cultured with various concentrations of CD (12.5, 25, and 50 μg/mL) and incubated for 48 h. The MTT assay showed that the treatment with CD did not induce cytotoxic effects on 3T3-L1 cells (Figure 6A). Therefore, we used these non-cytotoxic concentrations in subsequent in vitro studies. To confirm the inhibition of intracellular lipid accumulation by CD, 3T3-L1 cells were seeded in 12-well plates and various concentrations of CD (12.5, 25, and 50 μg/mL) were used. Microscopic images revealed that a high amount of lipids were stained in the control group (Figure 6B). However, lipid accumulation was reduced in the CD-treated groups. Particularly high lipid reduction was observed with high concentrations (25 and 50 μg/mL) of CD (Figure 6C). To elucidate the lipid inhibitory mechanism of CD, we analyzed the effect of CD on the expression of adipogenic proteins, such as sterol regulatory element-binding protein-1 (SREBP-1), peroxisome proliferator-activated receptor-γ (PPAR-γ), CCAAT/enhancer-binding protein-α (C/EBP-α), and fatty acid binding protein 4 (FABP4), by Western blotting. Figure 6D shows the Western blot bands of SREBP-1, PPAR-γ, C/EBP-α, and FABP4 in 3T3-L1 cells treated with 12.5, 25, and 50 μg/mL of CD. The intensity of adipogenic protein bands was calculated by the ImageJ program downloaded from NIH (http://imagej.nih.gov/ij/), which revealed that high concentration of CD dramatically inhibited the expression of SREBP-1, PPAR-γ, C/EBP-α, and FABP4 compared to the control group. The highest concentration of CD (50 μg/mL) suppressed the expression of adipogenic proteins SREBP-1 (0.37 ± 0.01 fold), PPAR-γ (0.55 ± 0.02 fold), C/EBP-α (0.51 ± 0.02 fold), and FABP4 (0.70 ± 0.05 fold) (Figure 6E). These results demonstrated that CD was not cytotoxic to 3T3-L1 cells and significantly lowered the reduction of lipid accumulation in differentiated 3T3-L1 cells via the downregulation of adipogenic protein expressions

## 3. Discussion

Obesity is a highly prevalent health hazard worldwide. It is characterized by high fat accumulation in adipocytes, high levels of lipids in the blood, and high blood pressure. Obesity is caused by complex interactions among the environment, genetics, and individual behavior. Thus, there is no single etiology for obesity. Many researchers have reported the epidemiology of obesity and have concluded that there are several trends, such as adult and childhood obesity in many countries [37,38,39]. However, there is a substantial lack of awareness as to the risk of obesity in the global population. Long-lasting overweight and obese conditions can induce metabolic and cardiovascular disease [40,41,42]. Thus, many researchers have tried to discover new chemicals that exhibit anti-obesity activity and are non-toxic to the human body. Recently, various natural products including polysaccharide, lipid, polyphenol, peptide, and pigment derived from marine bioresources have demonstrated anti-obesity activity [43] and a number of studies have investigated the potential anti-obesity effect of marine-derived natural products on adipogenic and thermogenic protein expressions [29,43,44,45,46,47,48,49,50]. In this study, the lipid inhibitory effect of *Grateloupia elliptica* (*G. elliptica*) 60% ethanol extract (GEE) was evaluated, and the GEE was fractionated using organic solvents (hexane, chloroform, ethyl acetate, buthanol) to isolate chlorophyll derivative (CD), and existing lipid lowering activity.

Previous publications reported that marine-derived carotenoids have various biological activities and health benefits effects [51,52,53,54]. In addition, the anti-obesity effect of carotenoids were comprehensively investigated [55]. Among them, fucoxanthin from brown seaweed was known as a popular bioactive component. Accordingly, there is a lot of anti-obesity research on fucoxanthin [56,57,58,59,60]. However, there are only a few reports which revealed the anti-obesity effects of CD. A previous publication reported that CD from the marine cyanobacteria showed a significant anti-obesity activity in vitro and in vivo studies [61]. Furthermore Seo et al. reported that chlorophyll from *Spirulina maxima* has potent lipid lowering activity via suppression of adipogenic markers, sterol regulatory element-binding protein-1 (SREBP-1), peroxisome proliferator-activated receptor-γ (PPAR-γ), CCAAT/enhancer-binding protein-α (C/EBP-α), fatty acid binding protein4 (FABP4), lysophosphatidic acid acryltransferase β (LPAATβ), acetyl CoA carboxylase (ACC), lipin1, fatty acid synthase (FAS), and diglyceride acryltransferase1 (DGAT1) in 3T3-L1 adipocytes [62].

Adipogenesis is controlled by the expression of specific adipogenic proteins that play an important role in fat accumulation in the adipocyte. Among the adipogenic proteins, SREBP-1, PPAR-γ, and C/EBP-α are crucial to adipocyte differentiation and fatty acid synthesis in adipocytes [63,64,65]. The function of FABP4 is the transportation of fatty acids between the extracellular matrix and intracellular matrix [66]. According to previous publications, the expression of SREBP-1 affects adipogenesis by regulating the expression of adipogenic proteins such as FAS, acetyl-CoA carboxylase, and FABP4 [67,68,69]. Jeon et al. reported that PPAR-γ and C/EBP-α are major transcription factors related to adipogenesis that induce transcriptional activation in the adipocyte. Furthermore, these two factors are synergistically activated by adipocyte-specific gene expression [70,71,72]. Therefore, suppressing the expression of adipogenic proteins is an effective strategy for inhibiting the differentiation and lipid accumulation of 3T3-L1 adipocytes.

In this study, the potential lipid inhibitory effect of red seaweeds was tested and *G. elliptica* was selected to isolate compounds exhibiting strong lipid inhibitory activity. Follow the isolation steps, CD were isolated from GEE and the lipid inhibitory activity of CD was evaluated in 3T3-L1 adipocytes. In our findings, 12.5, 25 and 50 μg/mL of CD dose did not have cytotoxic effects and highly reduced lipid accumulation in 3T3-L1 cells. Furthermore, Western blot analysis has shown that CD from GEE markedly reduced the adipogenic SREBP-1, PPAR-γ, C/EBP-α, and FABP4. These results corresponded to prior research [62], indicating that marine-derived chlorophylls could reduce lipid accumulation via downregulation of the major adipogenic protein expressions in 3T3-L1 cells. According to previous publications, there are some regulatory mechanism between adipogenic proteins. Holton et al. found that the major adipogenic C/EBP and PPAR expression is regulated by SREBPs, such as SREBP-1a, SREBP-1c, and SREBP-2 [73]. According to Fajas et al., the activation of PPAR-γ could regulate by SREBP transcription factors, including SREBP-1a, SREBP-1c, and SREBP-2, through PPAR-γ1 and PPAR-γ3 promotors [74], and Kim et al. also reported that SREBP-1 specifically increases PPAR-γ activity through the production of endogenous ligands of PPAR-γ [75]. Taken together, these results imply that the adipogenesis and lipid accumulation are predominantly regulated by SREBP-1. Accordingly, we suggested that the excellent lipid inhibitory effect of GEE may result from the suppression of adipogenic SREBP-1 expression in adipocytes.

In conclusion, GEE and its active compound CD showed strong suppression on intracellular lipid accumulation in 3T3-L1 adipocytes. These effects were achieved by the downregulation of adipogenic protein (SREBP-1, PPAR-γ, C/EBP-α, and FABP4) expressions. These results imply that CD is the active compound in GEE that induces the suppression of lipid accumulation. Taken together, our results demonstrate that chlorophyll derivative from GEE could be applied for medicinal and functional food purposes for the treatment of obesity.

## 4. Materials and Methods

### 4.1. Reagents

The Oil Red O (ORO) stain and MTT were purchased from Sigma Chemical Co. (St. Louis, MO, USA). The cell culture medium and supplements including Dulbecco’s Modified Eagle’s Medium (DMEM), Fetal Bovine Serum (FBS), HI Bovine Serum (BS), 0.5% Trypsin-EDTA, phosphate buffer saline (PBS) and Pen strep were purchased from GIBCO-BRL (Grand Island, NY, USA). Adipogenesis-related specific primary antibodies such as PPAR-γ, FABP4, and C/EBP-α were obtained from Cell Signaling Technology (Bedford, MA, USA). SREBP-1 and glyceraldehyde 3-phosphate dehydrogenase (GAPDH) were from Santa Cruz Biotechnology (Santa Cruz, CA, USA). Cell differentiation reagents including 3-isobutyl-1-methylxanthine (IBMX), dexamethasone, and insulin were purchased from Sigma Chemical Co. (St. Louis, MO, USA).

### 4.2. Extraction and Solvent Fractionation of 60% of Red Seaweeds Ethanol Extract

All the red seaweeds were collected from the Jeju Island of Korea. Then, the collected seaweed was washed with fresh water several times to remove salt, sand, and epiphytes, and stored at −80 °C in the deep freezer. Next, the frozen samples were freeze-dried and homogenized for ethanol extraction. Dried seaweed powder (100 g) was extracted with 1 L of 60% ethanol solution for 20 h under high temperature (70 °C) in a shaking incubator. After 20 h, the extracts were filtered and concentrated using a vacuum evaporator. The concentrated samples were referred to as ethanolic extracts of various red seaweeds.

In order to separate the active compounds, the solvent fractionation was performed following the modified method from Fernando et al. [76]. The ethanolic extract was suspended in distilled water and fractionated with four organic solvents including chloroform, ethyl acetate, n-hexane, and n-butanol. The solvent fractionation was followed by the sequential steps (hexane–chloroform–ethyl acetate–buthanol). The fractions were collected and concentrated using a vacuum rotary evaporator.

### 4.3. HPLC Analysis

The high-performance liquid chromatography (HPLC) was performed using Waters HPLC system (Waters, MA, USA) equipped with a 2998 photodiode array (PDA) detector, 2707 autosampler, and 515 HPLC pump. The C18 column (4.6 × 150 mm, sunfire C18, octadecyl-silica (ODS)) was used for separation and the chromatograms were detected in the wavelength range of 200 to 700 nm. For the isolation of active compound from GEE, solvent A (acetonitrile) was used as a mobile phase and solvent B (water) was used as a stationary phase. Then, the GEE was eluted using a gradient of solvent A and solvent B at a flow rate of 1 mL/min. The gradient methods were as follows: 0 min, 95% B, 5% A, 60 min, 0% B, 100% A, 70 min, 0% B, 100% A. The absorption spectrums were analyzed by photo diode array (PDA) detector using full ultraviolet (UV) scanning.

### 4.4. Cell Culture

The 3T3-L1 cells, preadipocytes, were purchased from the American Type Culture Collection (ATCC) and cultivated in DMEM media supplemented with 10% of bovine serum (BS) and 1% of antibiotics under the optimal cell culture conditions (5% CO_2_, 37 °C). The cell culture was done according to the previous report by Lee et al. [31]. To induce adipocyte differentiation, 3T3-L1 cells were incubated in DMEM medium supplemented with 10% of fetal bovine serum, 1% antibiotics and MDI (methylisobutylxanthine-dexamethasone-insulin) solution containing of IBMX (0.5 mM), dexamethasone (0.25 μM), and insulin (5 μg/mL) at day 0. After 2 days, further cell differentiation was induced through the addition of insulin (5 μg/mL) at day 2. After 8 days, the mature adipocytes were fixed with 10% formalin and the appearance of the accumulated lipids in 3T3-L1 cells was determined by Oil Red O staining at day 8. During the adipocyte differentiation, GEE was administered at two-day intervals at days 0, 2, 4, and 6, except for the control group.

### 4.5. Cytotoxicity

The cytotoxicity of GEE on 3T3-L1 cells was analyzed by MTT assay based on Reference [33]. The 3T3-L1 cells were seeded in 48-well plates at a density of 1 × 10^5^ cells/well and incubated with the different concentrations of CD (12.5, 25, 50 μg/mL) for 48 h. After 48 h, MTT (2 mg/mL) was added to each well and incubated for 3–4 h. After 3 h of incubation, the formed formazan crystals in the cell were quantified using a microplate reader at 540 nm.

### 4.6. Oil Red O Staining

The Oil Red O (ORO) staining was adopted to evaluate the potential lipid inhibitory activity on differentiated adipocytes according to the method from Cho et al. [34]. The 3T3-L1 cells were seeded in 12-well plates and cell differentiation was induced with MDI solutions for 8 days. During the cell differentiation, the different concentration-tested samples were treated at days 0, 2, 4, and 6, except for the control group. After 8 days, the differentiated 3T3-L1 cells were washed with PBS and fixed in 10% of formalin solution for 1 h. After 1 h, the cells were washed with 60% 2-propanol and dried at room temperature (25 °C). After drying, the 3T3-L1 cells were stained with 0.6% of Oil Red O (ORO) solution. The intracellular lipids were eluted with 60% 2-propanol and the optical intensity was measured using a microplate reader at 500 nm.

### 4.7. Western Blot Analysis

The inhibitory effect of CD on the expression of adipogenic proteins such as sterol regulatory element-binding protein-1 (SREBP-1), peroxisome proliferator-activated receptor-γ (PPAR-γ), CCAAT/enhancer-binding protein-α (C/EBP-α), and fatty acid binding protein 4 (FABP 4) were evaluated by Western blot analysis. The 3T3-L1 cells were lysed using lysis buffer containing (10 mg/mL Aprotinin, 5 mM ethylenediaminetetraacetic acid (EDTA), 10 mg/mL leupeptin, 10 mM Na_4_P_2_O_7_, 100 mM NaF, 2 mM Na_3_VO_4_, 1% NP-40, 1 mM phenylmethylsulfonyl fluoride (PMSF), 20 mM Tris) for 1 h. After 1 h, the lysates were clarified by centrifugation (4 °C, 12,902× *g*, 20 min). After centrifugation, the supernatant was collected to prepare Western blot samples. The protein contents of lysates were determined by bicinchoninic acid (BCA) protein assay kit (Thermoscientific, Catalog: 23225). The prepared loading samples were normalized to 40 μg/mL and mixed with 4× sodium dodecyl sulfate (SDS) sample loading buffer containing dithiothreitol (DTT). The mixture was subjected to sodium dodecyl sulfate polyacrylamide gel electrophoresis (SDS-PAGE) and electrophoresed. Then, the electrophoresed proteins were transferred onto a nitrocellulose membrane. The membranes were blocked in 5% blocking buffer (2.65 mM KCl, 137 mM NaCl, 5% Skim milk, 25 mM Tris-HCl, 0.05% Tween 20) for 2–3 h. After blocking, the membranes were incubated with primary antibody to the target protein at a dilution of 1:1000 overnight at 4 °C. Following incubation with primary antibody, the secondary antibody from mouse or rabbit was conjugated for 2 h. Then, the membranes were gently washed with 1x tris-buffered saline with 0.1% tween 20 detergent (TBST) and the images of protein band were detected using the Fusion Solo imaging system (Vilber Lourmat, France) after incubation with the enhanced chemiluminescence (ECL) Western blotting detection kit.

### 4.8. Statistical Analysis

All experiments were conducted in triplicate and all measurements were expressed as the mean ± standard deviations (SD). The difference of values was compared with one-way analysis of variance (ANOVA) followed by Tukey and Duncan’s test using SPSS software (SPSS 14, Chicago, Illinois). *p*-values (* *p* < 0.05, ** *p* < 0.01, and *** *p* < 0.001 as compared to the control group, ^#^
*p* < 0.05, ^##^
*p* < 0.01, and ^###^
*p* < 0.001 as compared to the control group) are considered significant.

## Figures and Tables

**Figure 1 marinedrugs-19-00091-f001:**
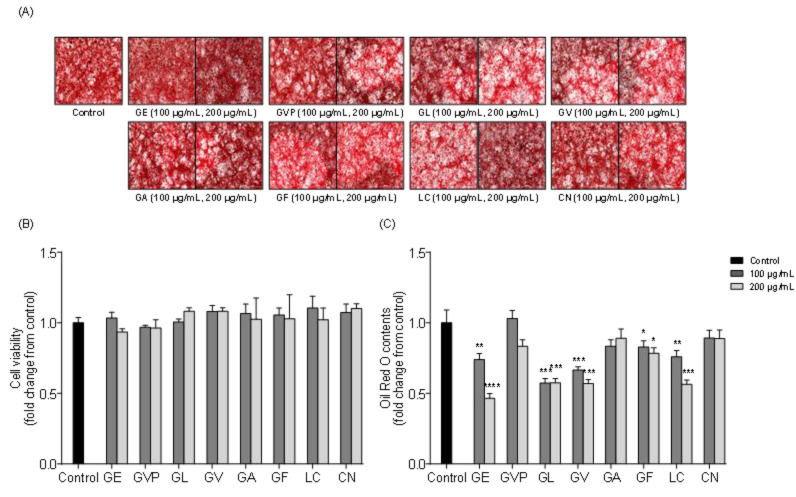
Inhibitory activity of red seaweed ethanol extracts on intracellular lipid accumulation during 3T3-L1 adipocytes differentiation: (**A**) Microscopic images of 3T3-L1 adipocyte as visualized by Oil Red O staining. (**B**) Cell viability by 3-(4,5-dimethylthiazol-2-yl)-2,5-diphenyltetrazolium (MTT) assay. (**C**) Relative lipid content was measured by microplate reader at 500 nm. Data are expressed as the means ± standard deviation (SD), n = 3 in each group. Significant difference was identified at * *p* < 0.05, ** *p* < 0.01, *** *p* < 0.001 and **** *p* < 0.0001 as compared to the control group. *Grateloupia elliptica* (GE), *Gracilaria vermiculophylla* (GVP), *Grateloupia lanceolata* (GL), *Gracilaria verrucosa* (GV), *Grateloupia asiatica* (GA), *Gloiopeltis furcata* (GF), *Lomentaria catenata* (LC), *Chondrus nipponicus* (CN).

**Figure 2 marinedrugs-19-00091-f002:**
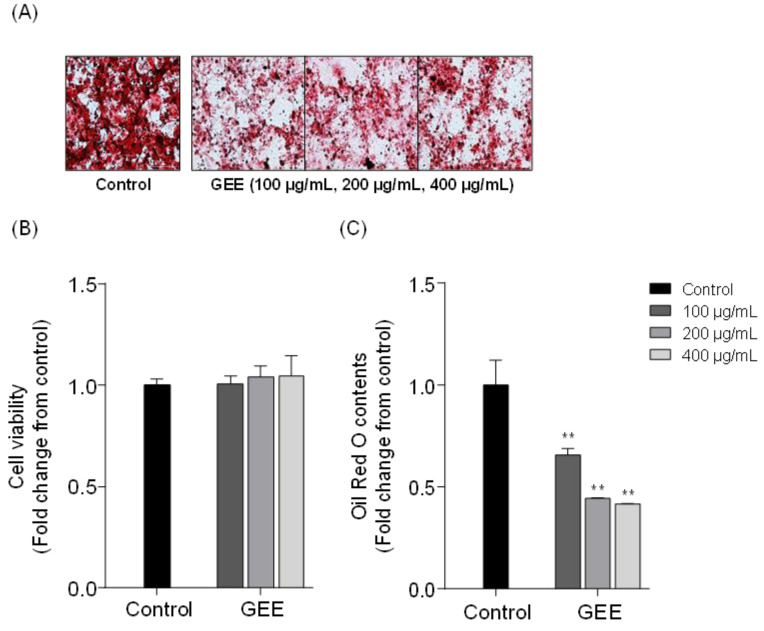
The effect of *Grateloupia elliptica* 60% ethanol extract (GEE) on intracellular lipid accumulation in 3T3-L1 adipocytes: (**A**) Microscopic images of 3T3-L1 adipocyte as visualized by Oil Red O staining. (**B**) Cell viability by MTT assay, (**C**) Relative lipid content measured by microplate reader at 500 nm. Data are expressed as the means ± SD, n = 3 in each group. Significant differences identified at ** *p* < 0.01 as compared to the control group.

**Figure 3 marinedrugs-19-00091-f003:**
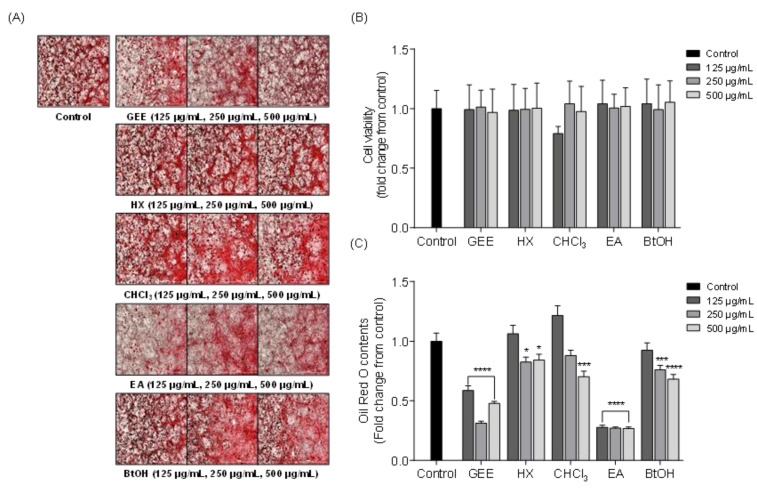
The effect of solvent fractions from GEE on intracellular lipid accumulation in 3T3-L1 adipocytes: (**A**) Microscopic images of 3T3-L1 adipocytes as visualized by Oil Red O staining. (**B**) Cell viability by MTT assay. (**C**) Relative lipid content was measured by microplate reader at 500 nm. Data are expressed as the means ± SD, n = 3 in each group. Significant difference was identified at * *p* < 0.05, *** *p* < 0.001 and **** *p* < 0.0001 as compared to the control group. *Grateloupia elliptica* 60% ethanol extract (GEE), hexane fraction (HX), chloroform fraction (CHCl_3_), ethyl acetate fraction (EA), butanol fraction (BtOH).

**Figure 4 marinedrugs-19-00091-f004:**
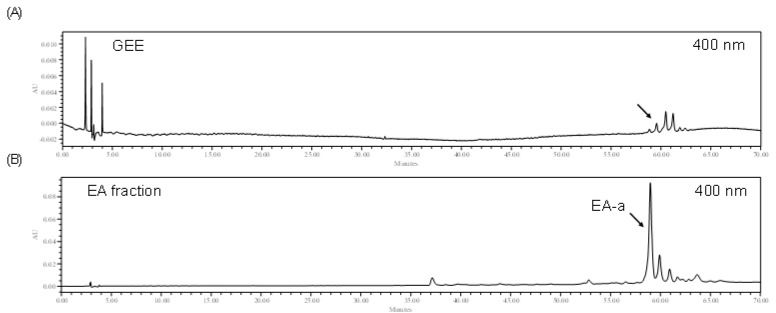
High-performance liquid chromatography (HPLC) analysis of GEE and its ethyl acetate (EA) fraction: (**A**) HPLC chromatogram of GEE, (**B**) HPLC chromatogram of EA fraction.

**Figure 5 marinedrugs-19-00091-f005:**
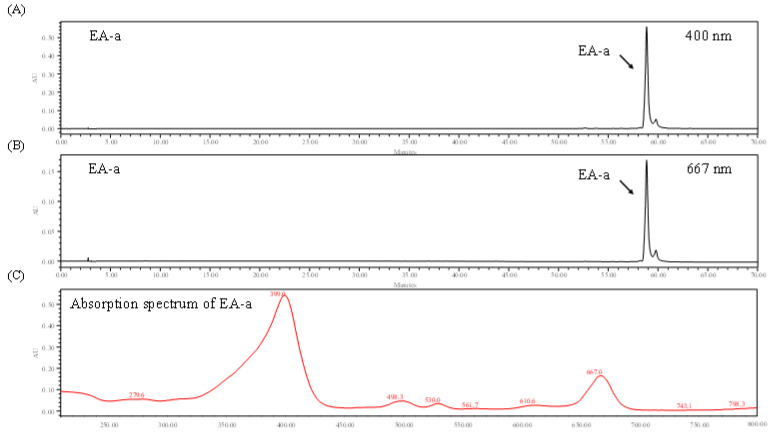
Identification and characterization of chlorophyll derivative (CD) from the EA fraction by HPLC chromatography: (**A**) HPLC chromatogram of EA-a at 400 nm, (**B**) HPLC chromatogram of EA-a at 667 nm, (**C**) absorption spectrum of EA-a.

**Figure 6 marinedrugs-19-00091-f006:**
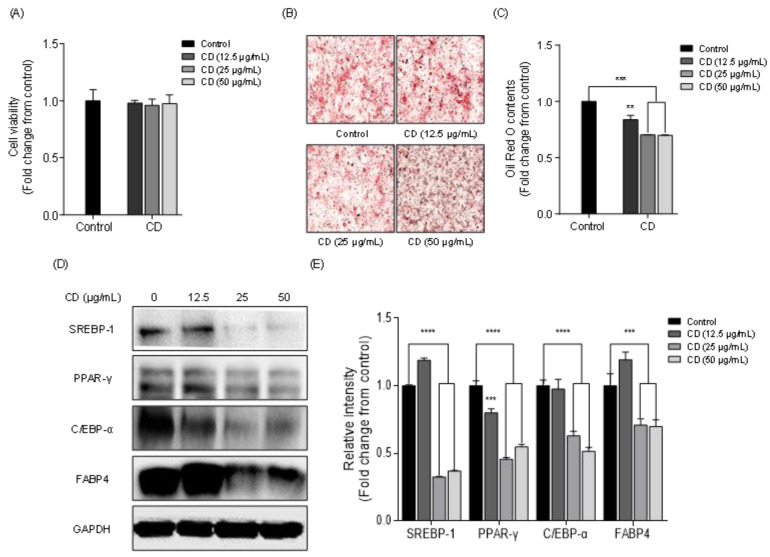
Effect of chlorophyll derivative (CD) on intracellular lipid accumulation and expression of adipogenic proteins in 3T3-L1 adipocytes: (**A**) MTT cytotoxicity assay, (**B**) microscopic images, (**C**) intracellular lipid content, (**D**) Western blot protein bands of adipogenic proteins (SREBP-1, PPAR- γ, C/EBP-α, FABP4) in 3T3-L1 adipocytes, and (**E**) comparative concentrations of adipogenic proteins isolated in Western blot as quantified by ImageJ. Data are expressed as the means ± SD, n = 3 in each group. Significant difference identified at ** *p* < 0.01, *** *p* < 0.001 and **** *p* < 0.0001 as compared to the control group. Sterol regulatory element-binding protein-1 (SREBP-1), peroxisome proliferator-activated receptor-γ (PPAR-γ), CCAAT/enhancer-binding protein-α (C/EBP-α), and fatty acid binding protein 4 (FABP4).

**Table 1 marinedrugs-19-00091-t001:** List of red seaweeds collected from Jeju Island, Korea.

No.	Scientific Name
1.	*Grateloupia elliptica* (GE)
2.	*Gracilaria vermiculophylla* (GVP)
3.	*Grateloupia lanceolata* (GL)
4.	*Gracilaria verrucosa* (GV)
5.	*Grateloupia asiatica* (GA)
6.	*Gloiopeltis furcata* (GF)
7.	*Lomentaria catenata* (LC)
8.	*Chondrus nipponicus* (CN)

## Data Availability

Not applicable.

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
