# Peer review of "Effects of Ethanol Extracts from Grateloupia elliptica, a Red Seaweed, and Its Chlorophyll Derivative on 3T3-L1 Adipocytes: Suppression of Lipid Accumulation through Downregulation of Adipogenic Protein Expression"

_marinedrugs, 2021, doi:10.3390/md19020091_

Round 1
Reviewer 1 Report
The present manuscript describes the effect of G. elliptica and the cholorophyll derivate as a new potential natural anti-obesity drug. The study is well designed, and the methodology applied is adequate. Likewise, most of the results are well detailed and support the conclusion obtained. However, there are several points that authors should be revised previously to be accept the manuscript:
Minor comment
- Title is too large and very descriptive. Please, include a new and short title and more adequate to the journal. Most examples were consulted in paper published in the journal web page.
- Line 304 there is a mistake “cell culture and differentiation”
Mayor comments
- Related to comparative red seaweeds study described in figure 1, the effect of other species such as GL and GV on ORO content should be better than GE, due to obtained better results at lower dosis (100 ug/ml) (Fig 1C). Moreover, the cell viability showed in figure 1B is also better than GEE. Have authors any explication to these effects? Is there any additional benefit of GEE vs the other species (GL and GV)?
- The quality of the ORO images are very poor. Please include new images with more resolution. Include also the magnification of the images in the figures and in the figure legend.
- Figure 1. Include the time exposition of the different treatments in figure legend. Is more exact include in the title “Inhibitory activity of red seaweed ethanol extract on intracellular lipid accumulation during 3T3-L1 adipocyte differentiation”
- No reference of different red seaweeds treatment and extraction were including in the method section. The extracts are obtained similar to G. elliptica? (same conditions?)
- Describe with more detail the 4 times of GEE during cell differentiation (Day 0,2,4,6,8…???). is this protocol similar to CD experiments? Please include it in the corresponding method section.
- Figure 2 does not provide additional information of interest with respect to Figure 1. Are differences between doses used (100 vs. 200 or 400 ug/mL? ) Please, include more information on doses and classical pharmacological parameters (doses-response curve, EC50, efficacy maxima….) to describe in more detail the dose-response effect of GEE or suppress the image.
Author Response
Journal name: Marine drugs
Manuscript ID: marinedrugs-11071284
We much appreciate the time and effort taken by the journal editor and all reviewers in reviewing our manuscript. We are delighted to receive this revision from the “Marine drugs”. The reviewer have given valuable insights to improve our manuscript. We have given our best to provide clarifications to reviewer comments and to modify our manuscript. It is our most sincere hope that this revised manuscript would meet the journal requirements for publication.
Comments and Suggestions for Authors
Comment (1): Related to comparative red seaweeds study described in figure 1, the effect of other species such as GL and GV on ORO content should be better than GE, due to obtained better results at lower dosis (100 ug/ml) (Fig 1C). Moreover, the cell viability showed in figure 1B is also better than GEE. Have authors any explication to these effects? Is there any additional benefit of GEE vs the other species (GL and GV)?
Response: Thank you for your comment. Actually, the ORO contents more higher in the low concentration GEE treated group than GL and GV group. However, the high concentration of GEE showed significant lipid inhibitory activity compared with that of GL and GV group. Additionally, in this study the GEE was chosen for target samples because it has significant lipid inhibitory activity and the biomass huge in jeju island, korea.
Comment (2): The quality of the ORO images are very poor. Please include new images with more resolution. Include also the magnification of the images in the figures and in the figure legend.
Response: Thank you for your comment and suggestion. In this study, we focused relative lipid accumulation in 3T3-L1cells. Therefore, we not included magnification in the photos. In addition, the ORO images were made by photoshop software but the quality of the image very poor so we try to increase the quality of photos.
Comment (3): Figure 1. Include the time exposition of the different treatments in figure legend. Is more exact include in the title “Inhibitory activity of red seaweed ethanol extract on intracellular lipid accumulation during 3T3-L1 adipocyte differentiation”
Response: We appreciate your suggestion. Following your suggestion, we revised this in the manuscript (Line 97-98).
Comment (4): No reference of different red seaweeds treatment and extraction were including in the method section. The extracts are obtained similar to G. elliptica? (same conditions?)
Response: Thank you for highlighting of this mistake. All the red seaweeds samples were collected from the Jeju Island of korea and extracted same conditions. The revised version recorded in the manuscript (Line 283-295).
Comment (5): Describe with more detail the 4 times of GEE during cell differentiation (Day 0,2,4,6,8…???). is this protocol similar to CD experiments? Please include it in the corresponding method section.
Response: We agree your comment. The ORO staining protocol is same. Therefore, we revised the Oil Red O staining protocols in method section (Line 326-335)
Comment (6): Figure 2 does not provide additional information of interest with respect to Figure 1. Are differences between doses used (100 vs. 200 or 400 ug/mL? ) Please, include more information on doses and classical pharmacological parameters (doses-response curve, EC50, efficacy maxima….) to describe in more detail the dose-response effect of GEE or suppress the image.
Response: We appreciate your comment and suggestion. In Figure 2, we double checked the potential lipid inhibitor activity of GEE in differentiated 3T3-L1 cells and we could found that the range of 100-200 µg/mL is critical concentration.
Thank you in advance for your cooperation. I look forward to receiving your kind response.
Sincerely
You-Jin Jeon
Reviewer 2 Report
In this work by Lee et al. the authors evaluate the effects of the red seaweed G. elliptica on intracellular lipid accumulation, an interesting property that adds to other previously documented benefits. They compared several seaweeds and produce extracts that are further characterized for their capacity of reducing lipid accumulation in cultured pre-adipocite cells.
The manuscript is well written, clear in the scopes and adequately introduced. The methods are clear and rational although their presentation should by improved a bit. The results appear consistent with the methods and the images proposed. The discussion seems adequate and proportionated. Unfortunately, the authors do not converge to a characterization of one or more compound, but this is not really relevant for the efficacy of the overall message.
The overall quality of images is very low. Probably the is an issue with image resolution in the pdf, but I suggest to use a resolution of at least 200-300 dpi because as now it is really hard to read and understant.
Major comments:
1) is it not clear to me whether the procedures (and amounts) described for extracting compounds from G. elliptica was applied to the remaining 7 red seaweds described in table 1. Was 60% ethanol extract always used? Please clarify.
2) in figure 1C several extracts do not display dose dependent effect (e.g. GL). How can this be explained? Is ug/ml the final (i.e. in the plate) concentration? Authors should comment on this.
3) it is not clear to me the procedure of oil red staining: after fixation for 1h, cells are washed with 60% 2- propanol, then treated with oil red and the lipids eluted with 2-propanol. Why lipids are still on the plate after the first wash?
4) similar to 2), why in figure 3 dose dependency is not respected? This does not vote up for a real (or at least not obvious) direct effect. Authors should comment on this. In addition, statistical symbolism (asterisks and lines) is misleading. I assume that "NS" means not significant, so better to omit asterisks rather than writing NS.
5) the resolution of figure 5 is really too low: I suppose it is a screen shot from the HPLC software, that is not acceptable.
6) the densitometry in figure 6 D-E is questionable. In D, GAPDH signals appear saturated (authors should clarify if they monitored saturations, since ImageJ just report the highest possible value depending on the color depth and does not clearly indicate saturation) as well as C/EBP-a and FABP4: this risk to compromise the statistics and interpretation of E. Please clarify.
Minor comments: (l=line)
l118: the lexical construct "all … have no" should be rephrased to "none...have"
l211: there is a comma to remove after CD)
l262: the term "realize" should bye replaced with "achieve"
l283: "was" is missing
l286: why the authors started from 100 kg seaweed and 1000 L extraction solution? This is rather uncommon in similar literature…
l295: the company name is Waters (capital W)
l312: "4 times pf GEE were treated": please rephrase.
l314: "10% of" : please remove "of"
l381: possibly a problem of my viewer, but I see 1x105, should be 1x105 I suppose…
l321: "quantified by multiplate reader". Please rephrase, using "absorbance"…
l344 to l348: as it is it seems that normalization (or, better, "equalization") was performed after the SDS PAGE run.
l350: this sentence was substantially repeated on th next sentence, please remove.
l363: "significantly" should be "significant"
figure 5: please revise the legend: "by UV absorption spectrum", but 667 nm is not UV…
Author Response
Journal name: Marine drugs
Manuscript ID: marinedrugs-11071284
Thank you very much for spending your valuable time in assessing our manuscript. We appreciate you detailed review and salient comments. We have carried out necessary modifications to the manuscript based on your comments. The changes are marked in red color within the manuscript.
Comments and Suggestions for Authors
Major comments
Comment (1): is it not clear to me whether the procedures (and amounts) described for extracting compounds from G. elliptica was applied to the remaining 7 red seaweds described in table 1. Was 60% ethanol extract always used? Please clarify.
Response: Thank you for your comment. The 60% ethanol extraction was selected for screening test considering the application in industrial purposes. Based on the screening test the GEE was chosen for purification step. Therefore, the extracting compound applied on GEE.
Comment (2): in figure 1C several extracts do not display dose dependent effect (e.g. GL). How can this be explained? Is ug/ml the final (i.e. in the plate) concentration? Authors should comment on this.
Response: We agreed your comment. In Figure 1C, several extracts dose not showing dose dependent effect. However, our findings demonstrated that the ranges of 100-200 µg/ml is critical concentration.
Comment (3): it is not clear to me the procedure of oil red staining: after fixation for 1h, cells are washed with 60% 2- propanol, then treated with oil red and the lipids eluted with 2-propanol. Why lipids are still on the plate after the first wash?
Response: Thank you for your comment. The 60% of 2-propanol solution was used for washing to remove the remaining formalin solution. However, the accumulated lipids not removed by 60% 2- propanol washing.
Comment (4): similar to 2), why in figure 3 dose dependency is not respected? This does not vote up for a real (or at least not obvious) direct effect. Authors should comment on this. In addition, statistical symbolism (asterisks and lines) is misleading. I assume that "NS" means not significant, so better to omit asterisks rather than writing NS.
Response: Thank you for your comment and suggestion. Following your suggestion we have changed in Figure 3 and we suggest that the below of 125 µg/ml of EA is critical concentration.
Comment (5): the resolution of figure 5 is really too low: I suppose it is a screen shot from the HPLC software, that is not acceptable.
Response: Thank you for your observation. We have changed in Figure 5.
Comment (6): the densitometry in figure 6 D-E is questionable. In D, GAPDH signals appear saturated (authors should clarify if they monitored saturations, since ImageJ just report the highest possible value depending on the color depth and does not clearly indicate saturation) as well as C/EBP-a and FABP4: this risk to compromise the statistics and interpretation of E. Please clarify.
Response: Thank you for highlighting of this mistake. The western bland of GAPDH not saturated when we quantified the protein using ImageJ software. This mistake occurred when we prepare western band photos.
Minor comments
Comment (1): l118: the lexical construct "all … have no" should be rephrased to "none...have"
Response: Thank you for your comment and suggestion. According to your comment, we have changed in the manuscript (Line 117-118).
Comment (2): l211: there is a comma to remove after CD)
Response: Thank you for pointing out this mistake. According to your comment, we removed comma in the manuscript (Line 211-212).
Comment (3): l262: the term "realize" should bye replaced with "achieve"
Response: Thank you for your comment. Following your comment, we revised in the manuscript (Line 262-263).
Comment (4): l283: "was" is missing
Response: Thank you for your observation. We revised in the manuscript (Line 283-284).
Comment (5): why the authors started from 100 kg seaweed and 1000 L extraction solution? This is rather uncommon in similar literature…
Response: We are sorry to confusing you. We have changed in the materials and methods section (Line 286-287).
Comment (6): l286: why the authors started from 100 kg seaweed and 1000 L extraction solution? This is rather uncommon in similar literature…
Response: Same comment with above. We have changed in the materials and methods section (Line 286-287).
Comment (7): l295: the company name is Waters (capital W)
Response: Thank you for highlighting this mistake. We have revised in the manuscript (Line 297-298).
Comment (8): l312: "4 times pf GEE were treated": please rephrase.
Response: Thank you for your comment and suggestion. This sentence may be confusing you therefore we rephrase this sentence in the manuscript. (Line 314-317).
Comment (9): l314: "10% of" : please remove "of"
Response: Thank you for your observation. We have revise in the manuscript (Line 315).
Comment (10): l381: possibly a problem of my viewer, but I see 1x105, should be 1x105 I suppose…
Response: We agreed your comment. We have changed in the manuscript (Line 320).
Comment (11): l321: "quantified by multiplate reader". Please rephrase, using "absorbance"…
Response: Thank you for your comment. Following your comment, we have changed in the manuscript (Line 323).
Comment (12): l344 to l348: as it is it seems that normalization (or, better, "equalization") was performed after the SDS PAGE run.
Response: We agreed your comment and suggestion. Following your comment, we have changed in the manuscript (Line 345-349).
Comment (13): l350: this sentence was substantially repeated on th next sentence, please remove.
Response: Thank you for your comment. According to your comment, we have removed in the manuscript.
Comment (14): l363: "significantly" should be "significant"
Response: Thank you for your observation. According to your comment. We have revised in the manuscript (Line 363-364).
Comment (15): figure 5: please revise the legend: "by UV absorption spectrum", but 667 nm is not UV…
Response: Thank you for pointing out this mistake. We agreed your comment. Therefore we revised all the mistake in the manuscript (Line 145, 146, 147-148, 149, 151).
Thank you in advance for your cooperation. I look forward to receiving your kind response.
Sincerely
You-Jin Jeon
Round 2
Reviewer 2 Report
all my suggestions were taken into account, I have no more concerns apart from the resolution of figure 4 (only fig 5 was updated, but they both showed the same problem).
Author Response
Journal name: Marine drugs
Manuscript ID: marinedrugs-11071284
Comments and Suggestions for Authors
Major comments
Comment: all my suggestions were taken into account, I have no more concerns apart from the resolution of figure 4 (only fig 5 was updated, but they both showed the same problem).
Response: We agreed your comment. We again check the resolution of Figure 4 and changed in manuscript.